# Recency and rarity effects in disambiguating the focus of utterance: A developmental study

**Reiki Kishimoto** [iD] *, **Kazuhide Hashiya**

Faculty of Human-Environment Studies, Kyushu University, Fukuoka, Japan

* kishimoto.r.k@gmail.com

**Data Availability Statement:** DATA and CODE AVAILABILITY With the exception of sharing identifiable information about the participants, we released all materials and code for analysis in GitHub (https://github.com/ReikiKishimoto/

## Abstract

While the communicator's intended referent in a conversation may not be immediately apparent, effective communication often overcomes this ambiguity. However, the specific mechanisms through which children use various cues to pinpoint the referent remain unclear. The communicator determines what is salient from the receiver's perspective. In return, the receiver identifies what the communicator identifies to be relevant for the receiver. The current study focused on two salient cues: rarity and recency, because rarity results in surprise and recency means a cue is more easily perceived and remembered. The current study investigated how adults and children aged 7–10 employ rarity and recency cues embedded in a series of events to clarify the referent intended by the communicator. Participants observed sequences comprising one rare and eight frequent events. An utterance, "Did you see that?" was presented at the end of each sequence, and participants identified the event(s) referred to by "that." Events that were rare and close to the utterance were more likely to be identified as the referents. Notably, the utilization of these cues differed between adults and children. For adults, the recency effect manifested gradually, with events closer to the utterance identified more frequently, and it exhibited an interaction with rarity. Among children, the recency effect was absolute, as the event closest to the utterance held a higher likelihood of being identified, and this effect was not influenced by rarity. Two additional conditions eliminated potential response biases and memory-related confounds. Our research suggests that school-age children are capable of disambiguating utterances by factoring in that the events they find salient are likely to be the communicator's focus. However, they are still in the process of developing reasoning skills similar to those of adults.

## Introduction

Humans use speech signals as one of major external communicative vehicles to transmit information. In principle, as described in the classic schema proposed by Shannon (1948) [1], the communicator packages their focus into their speech and the receiver decodes the duplicate of their speech into the communicator's intended referent. However, what information is packed and how it is transmitted is not always explicit. Since any social species cannot exchange their perspectives as long as they have independent minds, they rely on shared communicative rules [2, 3].

disambiguation_of_Ima-no-mita) and OSF (https://osf.io/kex98/)

**Funding:** This work was financially supported by JSPS KAKENHI, Japan, Grant Numbers 22KJ2371 to RK, 18K02461, 19H04431, 19H05591, 20H01763, and 22H04929 to KH. The funders had no role in study design, data collection and analysis, decision to publish, or preparation of the manuscript.

**Competing interests:** The authors have declared that no competing interests exist.

Referential assignment requires the receiver to use pragmatic reasoning to decode the signals of the communicator. The receiver does this by factoring what motivated the speaker to communicate based on communicative rules that make the participants aware of their sharing. Consider a situation where *Peter* and *Mary* are looking up at the sky. A shooting star is passing by, and *Peter* says, "look at that!". Mary attributes "that" to the most salient event, despite numerous other possibilities like the clouds, stars, or moon.

The shooting star is the most perceptually salient object in the cognitive environment for *Mary* and thus she applies her perception to *Peter*'s experience [4, 5]. However, it is unclear whether *Mary*'s decoding of "that" corresponds to what *Peter* is referring to. This fact means that receivers do not always succeed in decoding the communicator's focus. However, in most cases humans are not aware of this possibility.

Information worthy of one's attention or cognitive efforts is that which yields significant cognitive effects and allows inference that would not have been possible without it [6]. Humans disambiguate information by discerning what information a communicator would perceive worthy from a receiver's perspective. The communicator manipulates the receiver's attentional state to maximize their cognitive effects. The receiver is aware that this occurs, because the receiver can also stand in the communicator's shoes. As a result of this mutual process, the receiver attributes the communicator's utterance to the relevant objects by merging contextual cues and conversational background with their prior knowledge [7].

Individuals believe that they share an epistemic state when they experience a perceptual input together. They behave as if they were aware that the other individual perceives, processes or memorizes the particular event in a similar way to their own. Conversations take place in a context where subjective expectations, assumptions, or beliefs about objective facts play a role in interpretation, which is not limited to informational background about the immediate physical environment or the immediately preceding utterances [6].

If perception of a given event is shared inter-individually, the level of saliency of external information depends on how the communicator perceives it. For instance, for individuals with typical hearing, the sound of a doorbell has a higher saliency than the ticking of a clock. The sound of a doorbell is generally louder (indicated by its higher physical amplitude), occurs rarely, and signals important social cues such as someone's arrival. On the contrary, the ticking sound of a clock occurs constantly in the background and provides minimal information to listeners. It is the communicative norm not only that their interlocutors share the same sensory perception, but also that they share a common understanding of the secondary information that those original perceptions imply. Disambiguation is an output from a calculation of various factors. Previous studies such as Murakami and Hashiya (2014) [7] explored one single factor such as the effect of recency. Our focus here is how children calculate multiple factors.

The amount of information in each event can be measured by the level of entropy, which indicates the certainty of forthcoming occurrences and/or the surprise that occurs when something unexpected happens [8]. Surprisal is a post hoc measure of event expectancy, while entropy is a measure of uncertainty about potential outcomes of future events. In the context of lexical processing, humans took a longer time when processing lexical information with higher surprisal values [9]. Surprise depends on the extent to which an event contradicted the perceiver's beliefs about the physical causality or the other agent's mental state [10, 11].

By nature, surprising events happen less frequently than normal events (If surprising events happened frequently, they would have less entropy and be no longer "surprising."). However, rare events do not necessarily contradict physical causality or known facts. Human perceive consistency and expect a familiar event to recur because entropy decreases as the same event recurs repeatedly. Thus, their expectation is violated when a different event occurs [12]. When events occur infrequently, a surprising event stands out as unfamiliar. In particular, the first

time an unfamiliar event occurs, it is perceived as novel—that is, salient—and is consequently likely to become a topic [13]. A variety of species, including invertebrates, can learn how to navigate a novel situation [14] and integrate novel information with the facts they already know. This leads to inferences they would not have made before acquiring this new information. The human cognitive tendency to be biased toward surprising events aligns with the mathematical understanding that observing rare events yields more information than frequent events [1].

Recency is another factor that affects disambiguation. Children attribute an ambiguous question to the most temporally proximate utterance when no other salient cues, such as rarity, are available. This tendency increases over development [7, 15]. The next question here is how children use other salient cues and whether these cues are outweighed by temporal proximity.

A wide range of animals capable of learning a novel association are able to understand physical causality, which states that "an event is generated by a preceding event" [16]. This causality suggests that the object that is spatially/temporally closest to the moved object is the more probable cause of that movement [17]. This theoretical prediction is the foundation of psychological research, which states that behavior (effect) is generated by what was just experienced (cause) [11, 18]. However, the time range varies across contexts and experimental procedures, such as a memory task in which the participant's response is retrieved after a delayed period.

Another perspective supporting the hypothesis that recency plays a pivotal role in disambiguation is the observation that animals tend to remember recent events more vividly [19]. Humans exhibit a bias towards recent events and this bias is also seen with syntax processing [20]. The recency principle predicts that newly encountered words or phrases are associated with recently processed material in the syntactic representation.

This assumes that important information is more likely to have been accessed more recently compared to less significant information, which would likely have been encoded further back in time [21]. Short-term memory is susceptible to decay over extended periods, whereas long-term memory can be challenging to be accessed. Humans and several species monitor the strength of a memory trace associated with specific events and are generally aware that memory weakens over time [22]. Some species, including humans, will not continually access a particular memory trace if it does not contain informational value. Thus, automatic decay of a memory is not always just a function of time [23]. When considering the fragile nature of memories over time and the inherent causality of temporal distance, it is probable that an individual's behavior, emotions, or thoughts would be influenced more by recent events that are remembered vividly.

The current study explored the development of pragmatic reasoning to identify how a receiver would pinpoint what the communicator thinks is relevant for the receiver. The consequence of this is that the receiver is able to decipher the utterance from the communicator's perspective. We developed an experimental setup in which participants were required to disambiguate utterances based on the interaction between the saliency from the receiver's (participant's) perspective, and the estimated saliency from the communicator's perspective.

To delineate the above-mentioned factors that contribute to the disambiguation process, we selected a Japanese phrase "*Ima-no mita*? (Did you see that?)", which is a commonly used phrase in daily conversation. "*Ima-no*" means "what just happened", and "*mita*" is the past tense of "miru", which means "see", in English. In the current study we thus used the English translation "Did you see that?".

In the current study, adults and children were individually shown video clips, in which nine identical spots were lined up in single a row. From each spot, one by one, and in order from left to right, a monster appeared, performed specific action, and then disappeared. Eight out of nine monsters performed the same action, and one monster performed a different action from the rest. Immediately after the last monster disappeared, an audio-recorded utterance "Did

you see that?" was presented. The participant was asked to identify which monster(s) the utterance was referring to.

The nonverbal response of the participant was recorded through a touch panel. Participants used their finger to draw a rectangle over the spot(s) that he/she thought the utterance referred to. This interface enabled the participant to choose single or multiple spots with no restrictions for how the nine events are grouped. For example, if they perceived the entire sequence as a single event, they could draw a rectangle encompassing all nine spots. Likewise, they could draw nine separate rectangles around the nine spots.

Recency and rarity would affect disambiguation because these two factors are salient and because humans behave in accordance with the belief that others perceive events in the same way they do. Additionally, based on the results of Murakami and Hashiya (2014) [7], it is hypothesized that when a rare event occurred at any point during a sequence of events, the degree of recency of this rare event increased as it neared the point of the utterance. If participants gave recency the most consideration in triggering an utterance, they would choose the closest event to the utterance. If participants gave rarity the most important gravity in triggering an utterance, they would choose the rare event. If participants gave the interaction of these two factors the most consideration, they would select the rare event more frequently, if it was presented more recently.

Even when participants' responses converged upon a pattern when one rare event happened, we cannot eliminate the alternative possibility that participants chose objects they strongly remembered at the time of choice without taking the communicator's perspective. To examine these possibilities, participants were shown video clips featuring two rare events with different degrees of recency. This condition examined whether participants put more gravity on rare events which happened more recently if they disambiguated the utterance not only based on rarity but also on recency. In the final condition, participants were shown video clips in which two rare events happened in a sequence, and one of them happened after the utterance. If the participant recognizes that one of the rare events occurs after the past-tense utterance and understands that, even if they find it salient, the communicator has not experienced it at the time of the utterance, participants would likely exclude it as a potential referent.

No previous studies have tested the interaction of these two factors by systematically manipulating them. Thus, our study is novel. In Murakami & Hashiya (2014) [7], they explored their question by testing children's response to a target question. However, the question "Did you see that?" is a yes/no question and does not require participants to elaborate on what they saw. To know this, it requires a third-party to ask participants to pick which one the communicator was referring to. In the current study, we asked participants, "Which events do you think the communicator was referring to when the communicator asked "did you see that?" One study verbally exploring the understanding of false belief [24] found that first order recursion (You think the communicator said XX) emerges in Japanese children at six to seven years of age. A clear understanding of recursion is a pre-requisite for the current study; thus, for practical reasons, selected participants were seven to ten years of age.

Another reason why the current study chose children aged seven and older is that seven or eight years of age is a transitional phase in which children learn to perceive a common knowledge state with others. One study demonstrated that children aged six to seven perceived a shared knowledge with others only when they received minimal communicative cues [25]. Children aged six to eight develop the ability to recognize others' knowledge or ignorance, which may differ from their own. Between ages four and eight, the percentage of children who could correctly recognize others' knowledge states increased [26]. Thus, at eight, they are still in the process of the development in perceiving shared experiences. One aspect of the current study explores how children disambiguate an utterance based on their perceived shared

experience, which potentially leads to common ground. Because of these reasons, we tested children aged seven and older.

## Experiment

### Ethics statement

In accordance with the Declaration of Helsinki, the procedure was approved by the ethics committee of the Faculty of Human-Environment Studies at Kyushu University (No. 2022–006). The recruitment of adult participants started on July 9, 2022, and finished on August 3, 2022. The recruitment of child participants started on August 14, 2022, and finished on December 29, 2022. For participant recruitment, participants or the children's caregivers were informed of the experimental procedures, including that the sessions will be recorded. Before starting the experiment, participants or their caregivers (in the case of child participants) gave their written consent for their participation.

### General methods

**Participants.**   The current experimental design has an open-ended aspect, in the sense that the process of disambiguation does not lead to a single "correct" interpretation. We thus started in adults to explore whether and how rarity and recency would interact in their disambiguation process and then compared the results with the ones obtained from children.

We recruited thirty-six Kyushu University students and staff as adult participants (16 females, age M = 23.5 years, SD = 4.5, range = 19–40) and eighty-two children (45 girls, age M = 84 months, SD = 13.8, range = 84–130) from our database of children whose parents had expressed interest in participating in developmental studies (see raw data file in OSF (URL: https://osf.io/kex98/) for all participants ages). All participants were confirmed to have grown up in Japan and speak Japanese as their primary language. One additional child was excluded because he did not show clear responses in all trials of the memory task (explained below).

**Stimulus.**   An event was represented by a monster who completed a specific action. Thirty-four distinct actions (a monster played a musical instrument, ate something, etc.) were prepared. The experiment consisted of twenty-four types of trials (6 trials in Single-Rare-Event Condition, 9 trials in Double-Rare-Events Condition and 9 trials in Utterance In-between Double-Rare-Events Condition (hereafter UI Double-Rare-Events Condition)) and participants completed each type of trial once.

To control for the carry-over effect of actions, 24 types of trials were randomly split into halves during which each action was used only once. Within each group, the Single-Rare-Event condition used one action as frequent events and one action as a rare event. In contrast, both Double-Rare-Events and UI Double-Rare-Events Conditions used one action as frequent events and two actions as rare events. Thirty-four unique actions were needed for each group. Actions were randomly assigned to each type of trial. A second group of trials was produced using the same procedure.

To control any artefact of combination of action or trial types and any other factors, the same three sets (set 1, 2, 3) of twenty-four types of trials were created with randomly assigning actions. The list of combinations of actions and types of trials and all stimulus are found in OSF.

**Design.**   Adults completed a group of 12 trials within a single set. Before completing the second group of 12 trials within the set, adults were given a 10-minute break during which they were allowed to do anything. The break aimed to reduce carry-over effects. Due to their less enduring concentration, children only completed a randomly selected set of twelve trials from a group of any set. The procedures were identical to those used for the experiments with adults except that children were given one practice trial which came from another set of trials.

This trial was not used for analysis. The complete set for adults (and single group for children) was randomly assigned to participants. The order of groups was randomized in adults. The order of trials was randomized across participants.

At the end of every trial, participants completed the memory task in which participants were tested whether they remember the insertion timing of utterance correctly (*SI* for the Participant Response Protocol).

**Procedure.** Participants sat on a chair, facing a 51-inch monitor (KJ-55X8500E, SONY) displaying video clips. They held a 15.6-inch touch panel (Spkulia) in their hands to make choices (S2 Fig).

## Experiments with adult participants

### Single-Rare-Event Condition with adult participants

Before the task, adult participants were told to respond as they did in the practice session (*SI* for the Practice Session Procedure). Adults watched videos in which nine monsters appeared sequentially from No. 1 to No. 9 and performed specific actions and then disappeared (Fig 1A). The first three monsters performed the same action and were used for habituation, so that participants would expect the same event to occur and a different event would be perceived as rare. The insertion of the rare event varied between spots No. 4 and No. 9, and event timing was denoted as #-6, #-5, #-4, #-3, #-2, or #-1. Immediately after an event #-1, a recorded auditory stimulus "Did you see that?" was played. The touch panel presented an image in which nine monsters protruded from each spot. Participants were tasked with selecting which a spot or spots they thought were referred to. They pressed a decision button on the upper left to proceed to the memory task.

In the memory task, the monitor on the touch panel presented an identical image of the reference assignment task except for that two red arrows were overlaid between No. 6 and No. 7, and next to No. 9. Participants marked the arrow which indicated the position of utterance by same procedure as the reference assignment task. Participants pressed a decision button on the upper left to proceed to the next trial (*SI* for Participant Response Protocol and video. GitHub: https://github.com/ReikiKishimoto/disambiguation_of_Ima-no-mita). The trials where correct responses were given in the memory task were analyzed. The accuracy of the memory task is shown in Table 1.

### Results | Single-Rare-Event Condition with adult participants

All analysis in the current study was conducted in R version 4.4.1 [27] with the car package. For each participant, the three conditions were tested in a single experiment. However, as each experiment explored a different question, we analyzed them independently.

Visualizing the distribution of responses confirms that the majority of responses converged on rare or recent events (S4 Fig for details). It seems reasonable that participants were largely categorized into either those who selected a rare event or those who selected a recent event (rare-event timing #-1). Those who used other strategies such as selecting all events equally were categorized as others (S2–S7 Tables for other strategies observed in the current study). The number of participants who chose each strategy was treated as a dependent variable. The null hypothesis, which posits that the percentage of participants who choose a rare event is not different from the percentage who choose a recent event, was analyzed using a multinomial logistic model. The analysis was performed on the pooled data excluding rare events that occurred at #-1 (36 trials) because if the participant chose the rare event, it is unclear whether they chose it because it was rare or because it was recent. The results showed that the

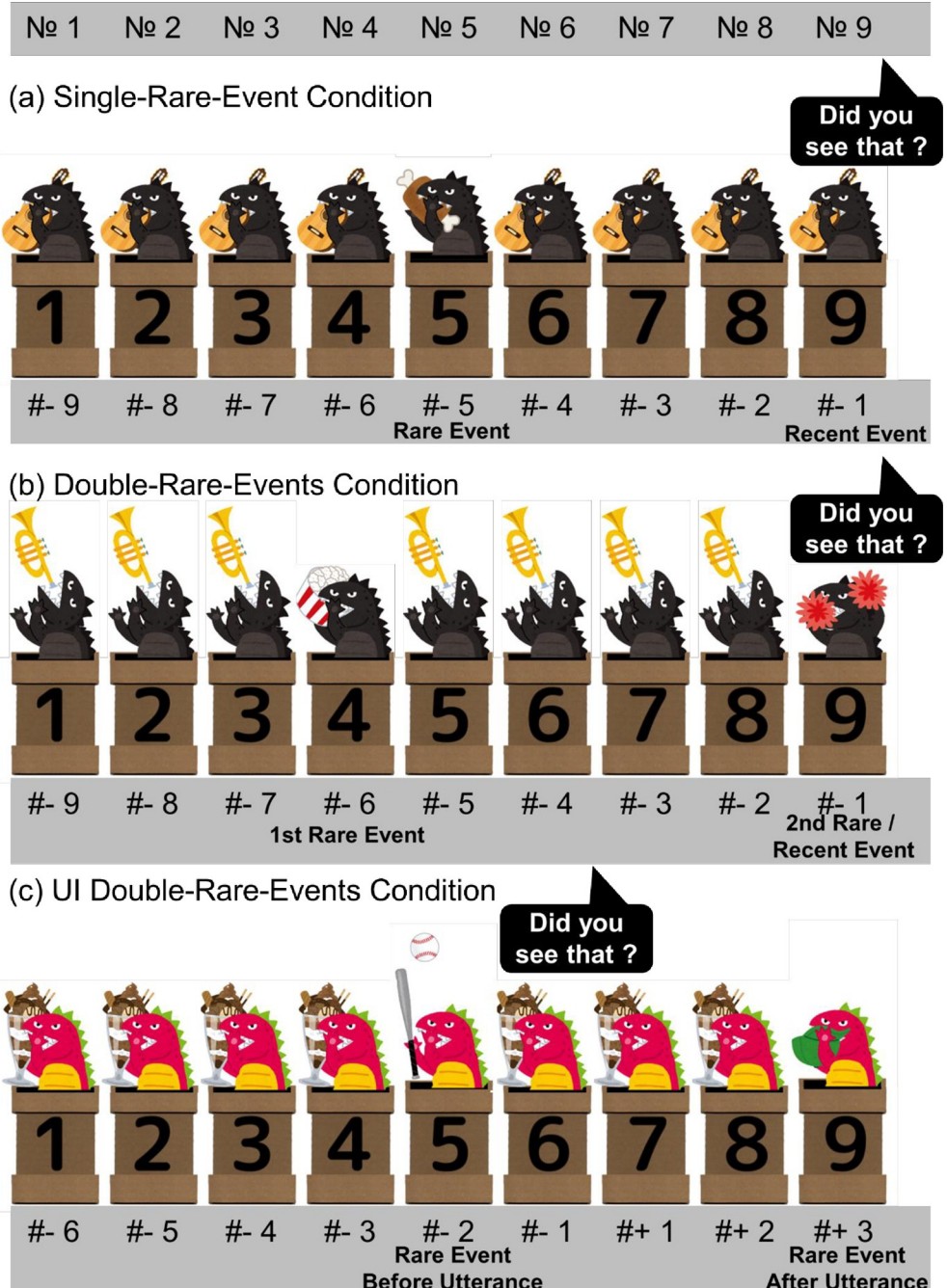

**Fig 1.** An example of the sequence of Single-Rare-Event Condition (a), Double Rare Events Condition (b) and UI Double Rare Events Condition (c), respectively. The numbers at the bottom denote the timing relative to the utterance. A negative value indicates a timing before the utterance, while a positive value signifies a timing after the utterance. Two kinds of monsters, black and red, were used alternately across trials so that participants would not lose interest. (a) An example of rare-event timing #-5. A monster emerged from a pipe, completed a specific action in 0.86s, and then disappeared back into the pipe. The next monster emerged after a 1-s interval. (b) An example of the rare-event timing #-6 / #-1. Two distinct actions were used for 1st rare event and 2nd rare event, respectively. (c) An example of the rare-event timing #-2 / #+3.

**Table 1. The accuracy of memory task in each condition.**

| Condition | Adults | Children |
|---|---|---|
| Single-Rare-Event | 215 / 216 (99.54%) | 233 / 246 (94.72%) |
| Double-Rare-Events | 324 / 324 (100%) | 346 / 368 (94.02%) |
| UI Double-Rare-Events | 318 / 324 (98.15%) | 356 / 370 (96.22%) |

percentage of time a rare event was chosen was significantly higher than that of recent events (vs. recent: z = -6.05, $p$ < .001) (right panel in Fig 2**A**).

To test whether participants randomly selected any of the nine events or they selected a recent event, the percentage of time a recent event (rare events that occurred at #-1) was chosen was compared to the chance level (right panel in Fig 2**A**). As a result, this percentage was significantly higher than chance (if participants randomly chose either a single or multiple event, the possible number of combinations is $2^9 - 1 = 511$ and the chance level is 1 / 511.; *SI* for the calculation). Our participants' strategies were divided into putting more weight on rarity and putting more weight on recency (105 vs. 30). At this point, it was unclear whether participants used the interaction of rarity and recency. Additionally, it was unclear whether they used these two factors in a separate manner or integrated them only when recency and rarity overlapped.

To test whether participants' choices were due to their response bias toward a rare event, participants were presented with two distinctive rare events within a single sequence, each differing in their recency; one being less recent (1st rare event) and another being more recent (2nd rare event). The fact that more participants used other strategies as rare events moved further from the utterance suggests that neither rarity nor recency was a reliable cue and they were unsure of a clear referent. It may be possible that participants assigned different weights to each event based on the degree of recency. This 2nd condition (Double-Rare-Events Condition) explored whether rarity and recency are interrelated with regard to disambiguation.

## Double-Rare-Events Condition with adult participants

The Double-Rare-Events Condition presented participants with an identical stimulus to the Single-Rare-Event Condition except that two distinct rare events happened. One of them occurred at any point between #-6 and #-4 and another occurred between #-3 and #-1, respectively (Fig 1**B**). This combination produced nine conditions in total.

## Results | Double-Rare-Events Condition with adult participants

The participants' responses converged on selecting one or two rare events or selecting a recent event (S5 Fig for details). The participants were categorized into four groups based on their selections: 2nd rare event (a rare event that occurred closer to the utterance), both rare events, a recent event, and other strategies. Selecting the 2nd rare event and selecting both rare events were treated as separate categories because the question here is to examine the interaction of recency and rarity. If participants perceived a 2nd rare event as the most salient due to the interaction between rarity and recency, the percentage of time a 2nd rare event was chosen was higher than in other categories. When rare events occurring at #-1 were excluded (108 trials), the percentage of time a 2nd rare event was chosen did not deviate from both rare events (z = 1.30, $p$ = .19) but was significantly higher than a recent event (z = -2.79, $p$ < .001). Unlike the Single-Rare-Event Condition where selecting a rare event that occurred at #-1 is akin to selecting a recent event, selecting a 2nd rare event that occurred at #-1 is not equivalent to selecting both rare events. Thus, rare events that occurred at #-1 were included in the analysis. The percentage of time a 2nd rare event was chosen was significantly higher than both rare

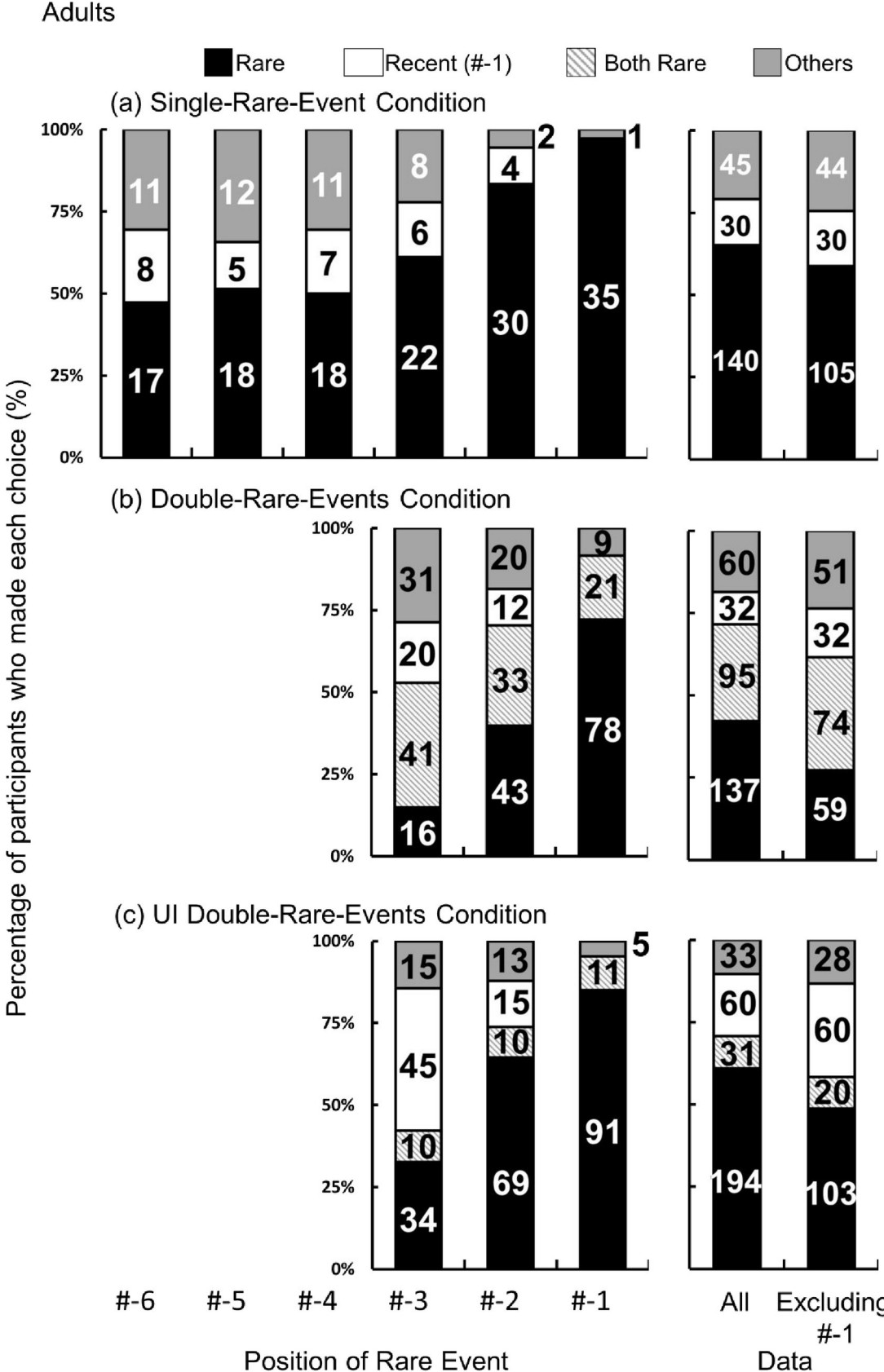

**Fig 2. The percentage of participant choices in adults.** The horizontal axis represents how far the rare event was from the utterance. The numbers in the bars indicate the number of participants that chose each option. **a** Single-Rare-Event Condition **b** Double-Rare-Events Condition **c** UI Double-Rare-Events Condition (b) The horizontal axis represents the position of a more recent rare event which occurred closer to the utterance.

events and a recent event (vs. both rare events: $z = -2.74$, $p < .01$; vs. recent event: $z = -7.41$, $p < .001$). Taken together, these results suggest that when rarity and recency overlapped, more participants selected only the 2nd rare event compared to both rare events. Adults chose only more recent rare events (2nd rare events) when two rare events happened, indicating that selection of a rare event observed in the Double-Rare-Events Condition as well as the Single-Rare-Event Condition was not likely due to a response bias for rare events.

The Double-Rare-Events Condition explored only a response bias for rare events. To explore the possibility of lacking the communicator's perspective, the third condition presented videos that were visually identical to those of the Double-Rare-Events Condition but the utterance occurred at a different timepoint such that a rare event occurred after the utterance. "Did you see that?" is a past tense phrase used to inquire about an experience of a past event and may be decoded for events that occurred before the utterance. If participants interpret the utterance by factoring into the interaction between the rarity and recency cues from integrated perspective that includes both their own and the communicator's viewpoints, they should not select a rare event that occurred after the utterance.

## UI Double-Rare-Events Condition with adult participants

To examine whether participants exclude a rare event occurring after a past tense phrase from potential referents, participants observed a sequence in which a rare event happened after the utterance. Participants observed another set of videos in which the communicator asked, "Did you see that?" immediately after the spot No. 6. A first rare event happened before the utterance between rare-event timing #-3 and #-1 while a second event happened after the utterance between rare-event timing #+1 and #+3, creating nine trials in total (Fig 1C). The videos were visually identical to those of the Double-Rare-Events Condition except that utterance was inserted after the spot No 6.

## Results| UI Double-Rare-Events Condition with adult participants

The participants response converged on choosing one or more rare events or choosing a recent event (S6 Fig for details). The participants were categorized based on their choices: a rare event, both rare events before and after the utterance, a recent event, and other strategies. The question here was whether participants omit a rare event after the utterance as a candidate for a referent. Thus, selecting both rare events and selecting only a rare event before the utterance were treated differently. Participants chose both rare events before and after the utterance more frequently than chance (1 / 511) (binomial, two-tailed, $p < .001$), even when they correctly demonstrated the timing at which the utterance was made in the memory task. The percentage of choosing both 1st and 2nd rare events (which is equivalent to rare events before and after the utterance in the UI Double-Rare-Events Condition) decreased in the UI Double-Rare-Events Condition compared to the Double-Rare-Events Condition (95 of 324 (29.32%) in Double-Rare-Events Condition vs. 31 of 318 (9.75%) in UI Double-Rare-Events Condition, $\chi2_{(1)} = 40.55$, $p < .001$, Odds ratio = 0.26). This result suggests that adults recognized the difference between these two conditions and employed distinct strategies for each. Participants were more prone to selecting events that captured their attention than theoretically expected. However, they were less likely to do so when an event occurred after the past tense utterance.

The null hypothesis was that the percentage of participants who choose a rare event before utterance is not different from that the percentage that choose both rare events before and after the utterance or a recent event. Thus, all conditions excluding rare events occurring at #-1(107 trials) were pooled into the analysis. The results showed that the percentage that chose a rare event before the utterance was significantly higher than the percentage that chose both rare events before and after utterance ($z = -6.70$, $p < .001$) or a recent event ($z = -3.33$, $p < .001$).

## Discussion

The three conditions tested investigated how adults allocated importance to aspects such as rarity, recency, the semantic meaning of "*Ima-no mita*?", and the timing of an utterance for pragmatic reasoning. The Single-Rare-Event Condition revealed that adults attributed the most importance to rarity. However, it remained unclear whether adults took all factors into account or relied only on rarity. The Double-Rare-Events Condition demonstrated that adults selected the most recent rare event most frequently when they observed two rare events within a sequence of events (and rarity and recency overlapped). The UI Double-Rare-Events Condition showed that compared to the theoretical baseline, adults chose rare events both before and after the utterance, even if they accurately recognized the position of the utterance. However, fewer adults chose rare events both before and after the utterance compared to when two rare events happened before the utterance (Double-Rare-Events Condition). Adults recognized the difference between these conditions and inhibited their response of selecting both rare events in UI Double-Rare-Events Condition.

## Experiments with children

### Analysis

The same analysis was applied for data collected in experiments with children.

### Results | Single-Rare-Event Condition with child participants

The same analysis used for the adults was performed to test the null hypothesis, which posits that the percentage of children that chose rare events is not different from those that chose recent events. The data excluding rare events occurring at #-1(39 trials) was analyzed. The results showed that the percentage of children that chose rare events significantly higher from that of recent events (vs. recent: $z = -3.32$, $p < .001$) and that the most frequently observed choice was a rare event (right panel in Fig 3A).

### Results | Double-Rare-Events Condition with child participants

We tested the null hypothesis that the percentage of children that chose a 2nd rare event was not different from those that chose both rare events or a recent event. Data except for 2nd rare events occurring at #-1 (116 trials) were pooled into the analysis. The results showed that the percentage children that chose a 2nd rare event was smaller than those that chose both rare events ($z = 2.11$, $p = .035$) and not different from the percentage children that chose a recent event ($z = -0.76$, $p = .44$) (right panel in Fig 3B). When rare events occurring at #-1 were included in the analysis, the percentage that chose a 2nd rare event was smaller than those that chose both rare events ($z = 3.58$, $p < .001$) a recent event ($z = 3.72$, $p < .001$). At this point, it is not clear whether the children did not use recency in disambiguation, they selected rare events as a response bias, or they chose any target that attracted their attention.

In the Double-Rare-Events Condition, children chose both rare events. On the contrary, in the Single-Rare-Event Condition, they were more likely to rely on recency. They seemed to

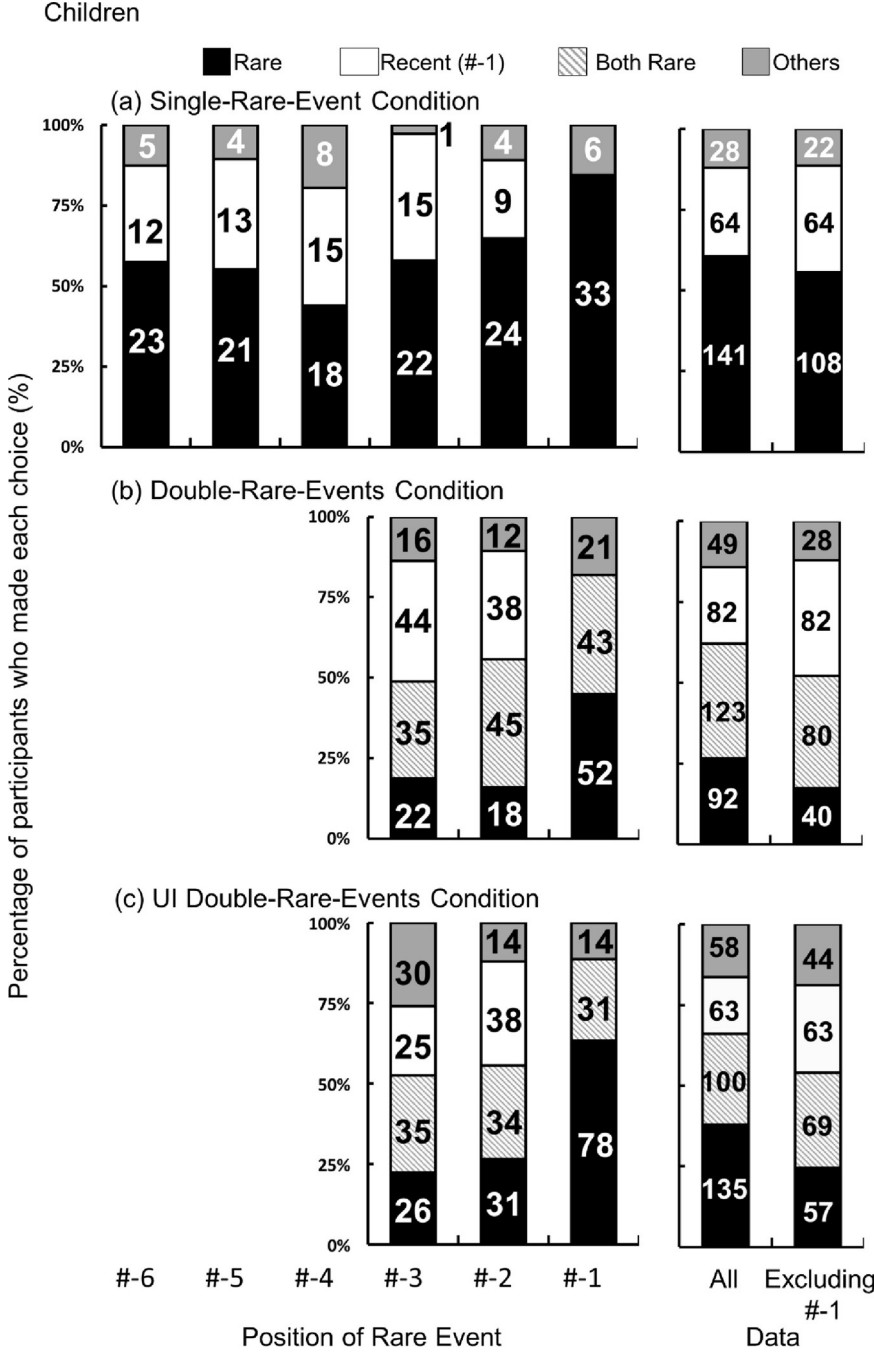

**Fig 3. The percentage of participant choices in children.** The number indicates the number of children who did each choice. The horizontal axis represents how far the rare event from the utterance. **a** Single-Rare-Event Condition **b** Double-Rare-Events Condition **c** UI Double-Rare-Events Condition (b) The horizontal axis represents the position of a more recent rare event which occurred closer to the utterance.

put more weight on rarity when the rare events happened more frequently and consequently, the rarity cue weakened. Children are sensitive to both recency and rarity, and which cue is more dominant is dependent on the degree of rarity.

## Results | UI Double-Rare-Event Condition with child participants

Participants chose both rare events before and after the utterance more frequently than chance (1 / 511) (binomial, two-tailed, $p < .001$), even when they correctly determined the timing of the utterance. The percentage choosing both 1st and 2nd rare events (which is equivalent to rare events before and after the utterance in the UI Double-Rare-Events Condition) decreased in the UI Double-Rare-Events Condition compared to the Double-Rare-Events Condition (123 of 346 (35.55%) in Double-Rare-Events Condition vs. 100 of 356 (28.09%) in UI Double-Rare-Events Condition, $\chi2_{(1)} = 4.51$, $p = .034$, Odds ratio = 0.71) (Right panels in Fig 3B and 3C), suggesting that children employed distinct strategies for each condition. While they were less likely to select events that preceded a past tense utterance, participants were more prone to selecting these events than the chance level when participants chose randomly. Again, children showed a rarity bias, which is consistent with the results of the Double-Rare-Events Condition. Children paid more attention to rare events and were less likely to factor in the communicator's perspective when making selections.

We tested the null hypothesis that the percentage of those that chose a rare event before the utterance was not different from those that chose both rare events or a recent event. Data except rare events occurring at #-1 (123trials) were pooled into the analysis. The results showed that the percentage of those that chose a rare event before an utterance did not deviate from those that chose both rare events ($z = 1.07$, $p = .29$) or a recent event ($z = 0.55$, $p = .58$) (Right panel in Fig 3C). Children did not show any specific preference for any strategy. They were divided into four groups based on where they placed more importance: (1) recency (Recent group in Fig 3), (2) rarity (Both Rare group in Fig 3), (3) the interaction of rarity, recency, and the communicator's perspective (Rare group in Fig 3) and (4) other strategies.

## Discussion

The Single-Rare-Event Condition revealed that children attributed the most importance to rarity, as adults did. The Double-Rare-Events Condition demonstrated that children selected both the 1st and 2nd rare events most frequently, indicating that their response observed in the Single-Rare-Event Condition may reflect their response bias towards rare events. The UI Double-Rare-Events Condition showed that children did not show any preference for a strategy. Even though children recognized the difference in the timing of utterance insertion between the Double-Rare-Events and UI Double-Rare-Events Conditions, they did not ignore a rare event that occurred after the past tense utterance. These results are in favor of the possibility that children selected rare events without considering other's perspective.

### Implications from comparing adults and children

To explore the developmental change, the data was compared between children and adults.

### Single-Rare-Event Condition

The percentage of those that chose a recent event was compared between adults and children to explore whether adults and children had the same inclination toward recency. The ratios of selecting a recent event (#-1) were compared between children and adults by logistic regression with the car and multicomp packages (Figs 2A and 3A) because the dependent variable is the number of respondents and not treated as continuous variables. All conditions except for condition #-1 were pooled and analyzed. The results showed that the ratio of selecting a recent event was significantly higher in children than in adults (30 of 179 (16.76%) in adults vs. 64 of

194 (32.99%) in children, $\chi^2_{(1)} = 13.273$, $p < .001$, Odds ratio = 2.45). Children selected a recent event more frequently than adults did.

Visual inspection of the data in the left panels of Figs 2 and 3 suggests that across adults and children, the percentage of participants that selected a rare event was flat until rare events occurring at #-4. For adults, the percentage increased as a rare event occurring at #-3 and approached the utterance. On the contrary, in children, the percentage remained flat until rare events occurring at #-2 and increased to the highest value for rare events occurring at #-1.

When a direction is clearly hypothesized, it can be expressed as a contrast [28]. This is the case for the current study because it also explored hypotheses generated through visual inspection. If a hypothesis predicts a particular pattern, this pattern can be represented numerically by assigning specific values to each condition. These assigned values, or contrast weights, capture the hypothesized differences in the means across experimental conditions. Thus, a trend analysis was performed for rare events occurring at #-3, #-2, and #-1. Two models were prepared: a linear model and a step model (Fig 4A and 4B; The rates of choosing a rare event follows pattern **a** in adults and pattern **b** in children). In the first linear model, the conditions were treated as a continuous variable and increased one by one. In the step model, the rare event percentage was different only in condition #-1. Therefore, we replaced #-3, #-2 and #-1 into -1, 0 ,1 in the linear model and -0.5, -0.5, 1 in the step model as a contrast coefficient,

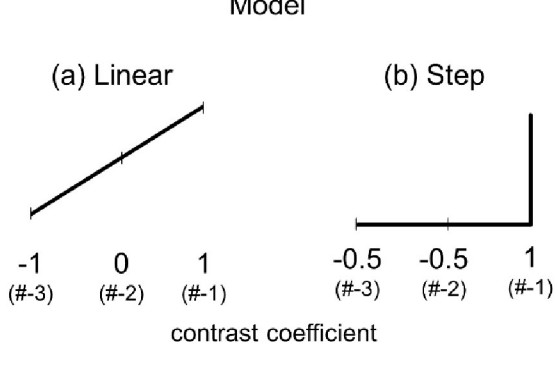

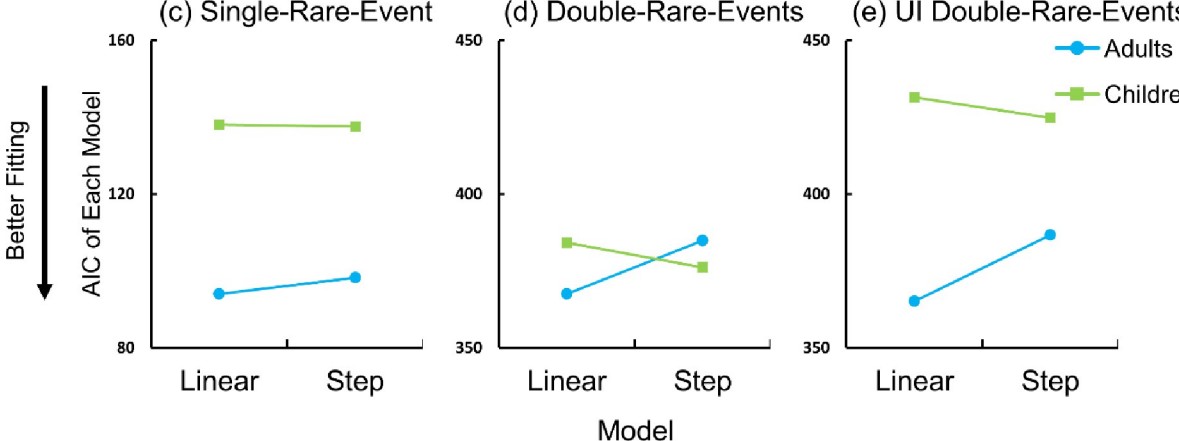

**Fig 4.** (a-b) The visualization of each prepared model. **a** The linear model assumes a linear increase as a function of a recency. **b** The step model assumes a discrete increase when recency peaks. The slopes are hypothetical to visualize the concept of analysis (b-d) The AICs of each condition. The y axis of the Single-Rare-Event Condition is different from the other two conditions due to the smaller sample size. **a** Single-Rare-Event Condition **b** Double-Rare-Events Condition **c** UI Double-Rare-Events Condition.

respectively. GLM (Generalized Linear Model) was performed on the percentage of choosing a rare event as a dependent variable by condition transformed into contrast coefficient.

AIC (Akaike's Information Criterion) is an indicator of a model's goodness of fit, and a lower value indicates a better fit. To determine whether each participant group's responses fit better to the linear or the step model, AICs for each model were compared. As a result of the analysis, adults and children showed inverted AIC trends for each model. In adults, the AIC was lower in the linear model than in the step model (94.028 vs. 98.22). On the contrary, in children, the AIC was higher in the linear model than in the step model (138 vs. 137.57) (Fig 4C). This result was supported by statistics, in favor of the hypothesis that adults' bias for rarity increased gradually as a rare event approached the utterance, whereas on the other hand, children selected a rare event occurring at #-1 most frequently.

The number of participants who selected a rare/recent event when a rare event occurred at #-1 is close to the sum of the participants who selected a rare or recent event when a rare event occurred at either #-3 or #-2. Thus, the fact that the highest percentage of participants selected a rare/recent event at #-1 might indicate that participants did not integrate recency and rarity and treated them separately. However, adults showed a gradual increase between #-3 and #-2, suggesting that they treated recency as a continuous factor when integrating with rarity. However, a limitation of our study is that it remains unclear whether children separated recency and rarity as categorical / stable factors, or whether they integrated them only when recency peaked.

## Double-Rare-Events Condition

The ratios of selecting a recent event (#-1) were compared between children and adults. All conditions excluding rare-event occurring at #-1 were pooled and analyzed. The results showed that the ratio of choosing a recent event was significantly higher in children than in adults (32 of 216 (14.81%) in adults vs. 82 of 230 (35.65%) in children, $\chi^2_{(1)}$ = 26.17, $p$ < .001, Odds ratio = 3.19). Children more frequently selected a recent event than adults did. This indicated that children's interpretation of the utterance was inclined to recency when the relative frequency of rare events increased to 2 / 9 from 1 / 9 (in the Single-Rare-Event Condition) and the effect of rarity was thus weakened.

The same trend analysis as in the Single-Rare-Event Condition was performed. Adults showed lower AIC for the linear model compared to the step model (367.44 vs. 384.93). Children showed higher AIC for linear model compared to the step model (384.15 vs. 376.1) (Fig 4D).

The number of adults who selected a rare/recent event when a rare event occurred at #-1 is lower than the total number of participants who selected a rare or recent event when a rare event occurred at either #-3 or #-2. This could be attributed to the fact that presenting two rare events had a stronger impact, leading some adults to choose both rare events.

## UI Double-Rare-Events Condition

The same analysis as in the Single-Rare-Event Condition was performed to test whether children chose a recent event more frequently than adults. All data except for condition #-1 was analyzed. Results showed that the ratios of selecting a recent event was not significantly different between adults and children (60 of 211 (28.44%) in adults vs. 63 of 233 (27.04%) in children, $\chi^2_{(1)}$ = 0.11, $p$ = .74; Odds ratio = 0.93). The difference between the first two conditions and the UI Double-Rare-Events Condition was that the relative ratio of frequent events became higher in this condition and consequently, a rare event became less infrequent (1/6 vs. 1/9). Participants watched six events which happened before the utterance (only one of which was rare) and thus the recency cue became stronger.

The same trend analysis was performed to test which model fitted better for adults and children, respectively. The AIC was lower in linear model than in step model in adults (365.07 vs. 386.68). Children showed an inversed tendency (431.52 vs. 424.82). Data from adults fitted to the linear model better and data from children fitted to the step model better (Fig 4E).

Again, the number of participants who selected a rare event when a rare event occurred at #-1 is close to the sum of the participants who selected a rare or recent event when a rare event occurred at either #-3 or #-2. Adults showed a linear increase as a function of position of the rare event. Thus, adults integrated rarity and recency. Children again treated recency either as categorical and separate or integrated them only when recency peaked.

## Discussion

In the Single-Rare-Event and Double-Rare-Events Conditions, children were more likely to select a recent event than adults. However, in the UI Double-Rare-Events Condition, the trend differed: adults were as equally inclined to select a recent event as the children. This difference could be attributed to the higher relative ratio of frequent events and stronger impact of recency in the UI Double-Rare-Events Condition. Children put more weight on recency signaled by clear expression of recency embedded in the utterance than adults.

Across the three conditions, adults showed a linear increase in selecting a rare event as it approached the utterance. In contrast, children showed a sharp increase in selecting a rare event when a rare event occurred right before the utterance. The results indicate that children treated recency as a categorical factor that does not vary, while adults treated recency as a continuous factor and integrated the gradient of recency with rarity.

## General discussion

This experiment, which tested three conditions, systematically explored whether and how participants used recency and rarity of a perceived event to disambiguate the utterance "Did you see that? ("*Ima-no mita*?" in Japanese)", and if the strategies differed between adults and children. The first (Single-Rare-Event) condition showed that adults and children alike used both rarity and recency as cues but depended more on rarity. The second (Double-Rare-Events) condition showed that adults distinguished two rare events by the degree of recency, though children selected two rare events regardless of the degree of recency. The third (UI Double-Rare-Events) condition confirmed that timing of the utterance insertion adequately influences the disambiguation process; not many adults or children choose the rare event after the utterance as referent. However, the results also suggested that more children selected the rare event after the utterance as referent, as compared to adults.

Children put more weight on the recency of an event compared to adults when sequences were longer, a rarity cue was stronger, or there were no other events that could have a stronger influence on participants' choice (such as rare events after the utterance). Adults showed a linear increase in the percentage of selecting a rare event whereas children showed a sharp increase only when rarity and recency existed together across the three conditions. Selecting a rare event when it occurred right before the utterance is equal to selecting a recent event. Given that the highest percentage of selecting a rare event is when it occurred most recently to the utterance, this means that recency and rarity cues synergize and are integrated. The evidence suggests that adults integrate rarity across the continuum of recency, while children treat recency as a categorical factor. However, a limitation of the current study is that it does not clarify why children most often selected a rare event when it occurred at position #-1. It could be that children integrate rarity and recency only when recency peaks, or it could be that

they consider rarity and recency independently. Based on the evidence, the strategy of disambiguation was different between adults and children as old as seven to ten years old.

In the current study, older children did not show the same patterns as adults. Children in this age range are still in the process of understanding others' degree of knowledge [25]. However, in this study, the dominant cues available were their own salient cues, viewed from the communicator's perspective. Research has shown that cognitive flexibility, especially in switching between tasks, has a longer developmental trajectory. Thirteen-year-old children have not yet reached adult levels [29]. Considering the recency gradient while integrating it with rarity requires greater cognitive flexibility than treating both factors as categorical. This might explain why the children in our study show patterns different from adults.

The utterance used in the current study was in the past tense, meaning events or objects that the communicator perceived prior to the utterance. Therefore, the reference should be to events that occurred before the utterance if the utterance is decoded in alignment with the semantic meaning. However, adults and children selected both rare events before and after the utterance (though not frequently) when one of the rare events occurred after the utterance. Moreover, this percentage surpassed the theoretical chance level. Adults and children correctly remembered that the utterance was made in the middle of the sequence. Thus, the possibility that participants mistakenly remembered that the utterance was made after the second rare event is unlikely.

The above-chance percentage of selecting a rare event after the utterance might have been due to an attentional bias that drew participants to aspects of a sequence that were salient from their perspective. However, another possibility is that participants may have predicted the communicator's upcoming utterance will be consistent with the preceding one. They might have had a belief that when the communicator referred to the first rare event, they might also be interested in a subsequent rare event and might refer to it if given the opportunity to elaborate on the utterance. Humans from the early stage of the development can extract an agent's intentions from their actions and can predict subsequent actions based on those intentions [30], even when an agent does not complete their intended actions [31–33]. The participants might over-interpret what the communicator might say in the future, predicting the communicator would act consistently with their initial action. However, this evidence is anecdotal and our interpretation remains speculative within the current paradigm. Future research should test these hypotheses regarding the underlying mechanisms.

Whereas children have a different strategy from adults, both adults and children disambiguate the utterance based on rarity, recency cues, or perceptual cues, at least from their own perspectives. In adults, P300 amplitudes in response to oddball stimuli increased as their probability decreased. Sensitivity to deviant stimuli has been observed in infants as young as 6 months [29], suggesting that attention to rare events develops early and persists into adulthood. The stimuli in the current study mirrored the oddball task structure, with a monster performing a different action serving as the oddball. Participants formed an expectation that the same event would recur, and when a rare event occurred, their expectation was violated, resulting in surprise. We developed our hypothesis based on the concept of entropy, surprise, memory, and causality. The results showed that participants across both age groups integrated these factors by attributing their perception to others. Children have already begun to understand that an element they perceive to be salient in a conversational context is likely to be perceived as salient by others as well. This belief allows them to pinpoint what is relevant for them from the communicator's perspective. In the current paradigm, participants and their communicative partner observed novel events together. Participants had a belief that they shared the same knowledge and epistemic state with the communicative partner and that the partner was also aware of this belief.

Infants teach information to others when they have privileged knowledge or access to it [34–36]. These kinds of teaching behaviors indicate that humans tend to form the common ground for further communication by compensating for the differences in knowledge and epistemic state of their communicative partner. Infants around 18 months of age perceive an event they previously shared with a communicative partner as more worthy of the partner's attention compared to other events that are familiar to the infants but not shared with the partner [37]. When the knowledge and epistemic state was shared, humans hypothesize the previously shared object or event is a relevant element for a partner. A subtle difference in the cognitive environment shared with the partner results in a varied response when it comes to directing the communicative partner's attention towards various aspects of the perceptual environment. In the current paradigm, the available cues were replaced by the receiver's subjective saliency, which was consequently adopted to the communicator's perspective.

In the current paradigm, participants and the communicative partner have the same degree of knowledge of the target event and mutually share this fact. A receiver should resolve a referential assignment by detecting which entities of their own memory a communicator wishes to direct their attention to. Understanding another person's attention is to understand that they have intentional control over their own perceptual environments [13, 38]. Receivers discern what aspect of the entity made the communicative partner perceive it was important enough to share with the receiver.

The earliest pragmatic reasoning—grounded in others' ignorance/knowledge or shared experiences—emerges at one year of age [13, 39]. Infants as young as one year old attribute others' emotional expressions such as "Wow!" or "Cool!" to objects that are new / unfamiliar to others [13]. These studies about disambiguation focused on whether infants perceive one factor: whether the target object was already known or not. The disambiguation in our daily life consists of a calculation of multiple factors: other's knowledge state, attentional state, epistemic state, perceptual saliency including rarity, recency, or physical amplitude and recent conversational background, etc. In an experimental setting where participants were asked explicit questions and then ambiguous questions, children aged three-to five updated their interpretation of an ambiguous utterance based on the preceding explicit utterance [7, 15]. However, these studies only dealt with recent conversational background and not with saliency of the stimulus presentations. The current study focused on how humans lead to an optimal solution by detecting relevance for another person's attention based on their calculation of multiple cues even when they are aware that their answer aligns with the communicator's intention. Adults integrated rarity and recency to disambiguate utterances whereas children have not yet started to integrate the rarity and the recency.

In fact, adult and child participants in the current study showed consistent patterns across conditions, respectively, suggesting that each age group shows a particular strategy. Specifically, adults might perceive "Did you see that?" as a polysemous utterance that varies depending on the communicator's interest, focus, perspective, experience or belief. In contrast, children interpret the same phrase more semantically, in other words, in a more literal manner. Children's interpretation of the demonstrative "that" might be restricted to the temporal regulation of the literal "that". This tendency may explain why children give more weight to recency, compared to adults.

The current study took the English translation, "Did you see that?" from the original Japanese phrase, "*Ima-no-mita*?". The setting of the current study assumes that the (virtual) communicator and participant shared the same visual experience at the time the utterance was made. The participant's task here should be to probe which part of this perceived common ground the communicator's interest was directed to. Considering that the communicative partners are attempting to form common ground, the critical focus depends on the

communicator's interest. When you are unsure of whether the partner also watched an event, the critical focus here is whether the partner watched or missed it. Thus, the more appropriate translation would be "Did you see that?". When you watched a recent movie and asked someone, "Did you see (meaning 'watch') that? the use of "you" means only the receiver because the communicator watched it previously and is not referring to a shared perceptual experience. Another example is when your friend says she ran into a TV celebrity and you ask, "Did you see him? In this case, you are primarily interested in the receiver's experience, which is distinct from your own. This form of questioning suggests an awareness that the communicator and receiver do not share the same experience.

An alternative translation of "Did we see that?" reflects a sense of shared perspective and inclusiveness. The use of "we" includes both the communicator and the receiver, offering a more communal way of addressing an observation or event and emphasizing that they shared the event. In contrast, the use of "you" includes only the receiver. Furthermore, the two forms of questioning suggest that individual perspectives are distinct from each other. When the communicator uses "Did you see that?", the communicator is interested in the other person's viewpoint but maintains a boundary between their own perspective and that of the receiver. On the other hand, when the communicator uses "Did we see that?", the question implies a collective perception through which an event or situation is viewed, aiming to integrate individual perspectives into a unified or shared understanding.

The current manuscript adopts "Did you see that?" as English translation for the original Japanese phrase used in the experiment. No subject (such as "you") do not appear in the original Japanese phrase "*Ima-no-mita*?". Deletion of subjects, objects, or other sentence elements is more commonly observed in Japanese as compared to English, where such deletions are strictly limited to the particular forms [40]. In English, ambiguity of a subject can also exist even when the subject is not omitted. As an example, "Did you see that?" may mean "Did we see that?", or "Did you see what I saw?". In the context of the current experimental setup, the participant shared visual experience with the communicator and interpreted her utterance based on that shared experience. The receiver must identify which aspect of the shared experience the communicator wants the receiver's attention directed to. In this sense, the participant must decipher "you" as "we". This perspective should be important for future studies in languages where the translation of "Did you see that?" may or may not include a subject.

When the communicator says, "Did you see that?" regarding the shared experience, his utterance is made on his assumption that you in fact saw it, and he intended to share his interest. Given the results of these three conditions, adults might decode the utterance based on a shared communicative norm, in other words, pragmatically; they might discern that the communicator intends to share surprising events that the communicator and receiver recently experienced together. On the other hand, children might put more weight on semantics. The current results cannot be explained by the difference in attentional bias between adults and children. The difference may be due to the difference in the decoding of popular "idiomatic" expressions. The current study clarified that children as old as seven to ten years showed a disambiguation strategy which is different from adults. At some point after the age of 10, humans may start to decode the ulterior meaning of the ambiguity for collaborative communication.

It remains an open question as to when children over the age of 10 learn the ability to disambiguate language in an adult-like manner. Ten-year-old children are still in the process of interpreting elements of utterances semantically. The current study was done with Japanese speakers. Future studies across a wider cultural or linguistic background including younger age exploring interaction between cultural and developmental effects would provide a new perspective to this area of study.

## Supporting information

**S1 File. Practice session procedure.**
(PDF)

**S2 File. Details of the procedure.**
(PDF)

**S3 File. Participant response protocol.**
(PDF)

**S4 File. Calculating the probability of each choice.**
(PDF)

**S5 File. The translation of the introduction video.**
(PDF)

**S6 File. The details for other strategies.**
(PDF)

**S1 Table. The sequence of the practice session.**
(DOCX)

**S2 Table. The categorization of other selection strategies in adults in the Single Rare-Event Condition.**
(DOCX)

**S3 Table. The categorization of other selection strategies in adults in the Double-Rare-Events Condition.**
(DOCX)

**S4 Table. The categorization of other selection strategies in adults in the UI Double-Rare-Events Condition.**
(DOCX)

**S5 Table. The categorization of other selection strategies in children in the Single-Rare-Event Condition.**
(DOCX)

**S6 Table. The categorization of other selection strategies in children in the Double-Rare-Events Condition.**
(DOCX)

**S7 Table. The categorization of other selection strategies in children in the UI Double-Rare-Events Condition.**
(DOCX)

**S1 Fig. An example of a practice session.** Each animal appeared with an accompanying sound indicating cat or dog. Each animal wore a different colored ribbon so that they could easily be differentiated.
(DOCX)

**S2 Fig. Schematic representation of the image on the monitor and touch screen.**
(DOCX)

**S3 Fig.** Distribution of response in (a) in Single-Rare-Event Condition, (b)Double-Rare-Events Condition and (c) UI Double-Rare-Events Condition, respectively.
(DOCX)

**S4 Fig. The distributions of responses for each option in the Single-Rare-Event Condition.**
(DOCX)

**S5 Fig. The distributions of responses on each option in the UI Double-Rare-Events Condition.**
(DOCX)

**S6 Fig. The distributions of responses on each option in the UI Double-Rare-Events Condition.**
(DOCX)

## Acknowledgments

This information is inserted in the submission system.

## Author Contributions

**Conceptualization:** Reiki Kishimoto, Kazuhide Hashiya.

**Data curation:** Reiki Kishimoto.

**Formal analysis:** Reiki Kishimoto.

**Funding acquisition:** Reiki Kishimoto, Kazuhide Hashiya.

**Investigation:** Reiki Kishimoto.

**Methodology:** Reiki Kishimoto, Kazuhide Hashiya.

**Project administration:** Kazuhide Hashiya.

**Resources:** Kazuhide Hashiya.

**Software:** Reiki Kishimoto.

**Supervision:** Kazuhide Hashiya.

**Validation:** Reiki Kishimoto.

**Visualization:** Reiki Kishimoto.

**Writing – original draft:** Reiki Kishimoto, Kazuhide Hashiya.

**Writing – review & editing:** Reiki Kishimoto, Kazuhide Hashiya.

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
