## [Decision Letter · Decision Letter 0]

4 Oct 2024

PONE-D-24-35473Recency and rarity effects in disambiguating the focus of utterance: A developmental studyPLOS ONE

Dear Dr. Kishimoto,

Thank you for submitting your manuscript to PLOS ONE. After careful consideration, we feel that it has merit but does not fully meet PLOS ONE’s publication criteria as it currently stands. Therefore, we invite you to submit a revised version of the manuscript that addresses the points raised during the review process.

The study reported in this manuscript is interesting and potentially meritorious. However, there are several major aspects of the manuscript that require revision before it can be considered for publication, as outlined in the reviewers' comments.  First, the Discussion should be aligned with the theoretical framework introduced in the Introduction. Second, theoretical justification for why the age group studied was selected is needed. Third, greater clarification of the coding scheme an its application to the data is needed. Finally, justification is needed for employing the analytic technique employed. 

We look forward to receiving your revised manuscript.

Kind regards,

Laura Morett

Academic Editor

PLOS ONE

Journal Requirements: When submitting your revision, we need you to address these additional requirements. 1. Please ensure that your manuscript meets PLOS ONE's style requirements, including those for file naming. The PLOS ONE style templates can be found at https://journals.plos.org/plosone/s/file?id=wjVg/PLOSOne_formatting_sample_main_body.pdf and https://journals.plos.org/plosone/s/file?id=ba62/PLOSOne_formatting_sample_title_authors_affiliations.pdf 2. Thank you for stating the following financial disclosure: "This work was financially supported by JSPS KAKENHI, Japan, Grant Number No. 21J00124 to RK, 18K02461, 19H04431, 19H05591, 20H01763, and 22H04929 to KH from the Japan Society for the Promotion of Science." Please state what role the funders took in the study.  If the funders had no role, please state: ""The funders had no role in study design, data collection and analysis, decision to publish, or preparation of the manuscript."" If this statement is not correct you must amend it as needed. Please include this amended Role of Funder statement in your cover letter; we will change the online submission form on your behalf. 3. Please expand the acronym “JSPS KAKENHI” (as indicated in your financial disclosure) so that it states the name of your funders in full. This information should be included in your cover letter; we will change the online submission form on your behalf. 4. Your ethics statement should only appear in the Methods section of your manuscript. If your ethics statement is written in any section besides the Methods, please move it to the Methods section and delete it from any other section. Please ensure that your ethics statement is included in your manuscript, as the ethics statement entered into the online submission form will not be published alongside your manuscript. 5. Please include a separate caption for each figure in your manuscript. 6. Please include captions for your Supporting Information files at the end of your manuscript, and update any in-text citations to match accordingly. Please see our Supporting Information guidelines for more information: http://journals.plos.org/plosone/s/supporting-information.

**Additional Editor Comments:**

The study reported in this manuscript is interesting and potentially meritorious. However, there are several major aspects of the manuscript that require revision before it can be considered for publication, as outlined in the reviewers' comments. First, the Discussion should be aligned with the theoretical framework introduced in the Introduction. Second, theoretical justification for why the age group studied was selected is needed. Third, greater clarification of the coding scheme an its application to the data is needed. Finally, justification is needed for employing the analytic technique employed.

Reviewers' comments:

Reviewer's Responses to Questions

**Comments to the Author**

1. Is the manuscript technically sound, and do the data support the conclusions?

Reviewer #1: Yes

Reviewer #2: Yes

2. Has the statistical analysis been performed appropriately and rigorously? 

Reviewer #1: Yes

Reviewer #2: I Don't Know

3. Have the authors made all data underlying the findings in their manuscript fully available?

Reviewer #1: Yes

Reviewer #2: Yes

4. Is the manuscript presented in an intelligible fashion and written in standard English?

Reviewer #1: Yes

Reviewer #2: Yes

5. Review Comments to the Author

Reviewer #1: I read through the reviewers’ comments and the authors’ responses in addition to the manuscript, and overall, I think the authors responded seriously to the reviewers’ comments and appropriately revised the manuscript. I have some comments to improve this manuscript.

Major points

I think Coding scheme (categorization of the participants’ responses) must be stated in detail. I understand that the participants put “rectangles” according to their interpretation of the communicator. This is a nice way to collect the participants’ interpretations because the participants can express their interpretation with more freedom than forced choice method. However, the categorization method sounds ambiguous.

Rectangles participants indicated may sometimes include more than one spot, such as one “rare” spot and one or two preceding spots (this may mean that “a change” occurred, from rare to not rare). How did you code such performance? I think it is necessary to describe the coding scheme completely in the text.

The results indicate that children treated recency and rarity cues separately, while adults treated recency as a continuous factor and integrated the gradient of recency with rarity.

I’m not quite convinced about the authors’ statement. How do you know about being “separate” and being “continuous”? You seem to state it based on the AIC results, but still not very clear to me. More explanation is needed.

Minor points

UII Double-Rare-Events

What does UII stand for? Could you tell me what kind of abbreviation (if it is an abbreviation) it is?

Sometimes one sentence is too long and difficult to read. Please state the content more

compact and concisely. For example,

The percentage of choosing both

404 1st and 2nd rare events (which is equivalent to rare events before and after the

405 utterance in the UII Double-Rare-Events Condition) decreased in the UII Double-Rare-

406 Events Condition compared to the Double-Rare-Events Condition (95 of 324 (29.32%)

407 in Double-Rare-Events Condition vs. 31 of 318 (9.75%) in UII Double-Rare-Events

Condition, χ2(1) 408 = 40.55, p < .001, Odds ratio = 0.26), suggesting that adults recognized

409 the difference between these two conditions and employed distinct strategies for each.

In the Supplementary material

“… that two read arrows were overlaid” -> should be “… that two red arrows were overlaid”

446the percentage of people 　　-> the percentage of children　

In the text, sometimes “children” and “people” seem to be mixed. “Children” sounds better.

Figure ２　　The vertical axis represents how far the 4 rare event from the utterance.

“Vertical “ axis -> should be “horizontal” axis, I think.

In the “Data of the figures, circumflex or hut　^-#1　is difficult to understand. What is the difference between -#1 and ^-#1 ? I thought that it means “excluding -#1 performance.” Is this correct? Anyway, please explain the meaning of ^-#1, and/or more understandable description of it in the figure is desirable.

In Figure ４　data points for adults and children look similar (with a slight color difference only)

In the Double-rare condition, 　only -1,-2,-３ are described. How about -4, -5, -6? In my understanding, these distances might have existed according to the illustration of the stimuli.

Reviewer #2: This paper systematically investigates the pragmatic processes involved in identifying the referent of the utterance "Did you see that?" from the perspectives of recency and rarity. The study includes three experimental conditions and compares behavioral data between children aged 7-10 and adults. The findings, which demonstrate how salience within a context shifts according to combinations of recency and rarity, provide intriguing insights into the underlying cognitive mechanisms.

However, there are several aspects that require further elaboration or clarification, which I will outline below.

・In the Introduction, the use of terms such as "epistemic state" and "entropy" provides a theoretical framework for understanding the cognitive processes under investigation. However, these terms are scarcely referenced in the Discussion section. Instead, the Discussion primarily focuses on "common ground" and the concept of "we," which creates a disconnect between the Introduction and the Discussion. It would improve clarity if the Discussion were more closely aligned with the Introduction, integrating the theoretical terms initially introduced to maintain coherence throughout the argument.

・The studies cited in the previous studies focus on infancy and early childhood, yet this study targets children aged 7-10. It is essential to clarify why this age group was selected and to provide a theoretical justification for its significance in the context of development. Specifically, discussing how this age range might represent a transitional phase in the development of pragmatic skills related to referential disambiguation would strengthen the theoretical positioning of the study.

・Regarding the results, the use of the trend analysis as a method to assume a model is not sufficiently justified. A more explicit rationale for employing this method, including a discussion of why it is particularly suitable for the current dataset and the types of patterns it aims to uncover, would be beneficial.

・Line 707: The description that "the communicator and participant had common ground" requires further clarification. While the description of "a shared visual experience at the time the utterance was made" is understandable, the current experiment appears to involve only a retrospective interpretation of the referent "that" in the utterance "Did you see that?" This raises doubts about whether such an interaction can truly be characterized as being based on common ground, especially given that no communicative behavior was observed between the experimenter and participant to explicitly establish this shared understanding during rare event.

・Furthermore, the discussion about "we" is intriguing, particularly the idea that a referential frame of "we" emerges when both the experimenter and participant are included in the events. However, it would be more effective to separate this point from the discussion of the disambiguation of "that" and the concept of "we-ness." Doing so would allow for a clearer examination of each concept's contribution to the overall findings.

6. PLOS authors have the option to publish the peer review history of their article (what does this mean?). If published, this will include your full peer review and any attached files.

Reviewer #1: **Yes: **Harumi Kobayashi

Reviewer #2: No

---

## [Author Response · Author response to Decision Letter 0]

18 Nov 2024

Dear Laura Morett

Dear professor Laura Morett

** November 2024

Thank you for inviting us to submit a revised version of our manuscript: “Recency and rarity effects in disambiguating the focus of utterance: A developmental study” (ID: PONE-D-24-35473) to PLOS ONE. We appreciate your time and effort as Editor and thank the two reviewers for providing insightful and valuable feedback. It is our great pleasure to resubmit our article for further consideration. We have incorporated changes that reflect the detailed suggestions you have raised. We also hope that our edits and responses satisfy all the issues and concerns that arose during the review process. We have highlighted the revised parts in our manuscript and present a point-by-point response to the questions and comments below.

Sincerely,

Reiki Kishimoto

Faculty of Human Environment Studies

Kyushu University

E-A-306, 744 Motooka, Nishi-ku, Fukuoka

819-0395, Japan 

E-mail: kishimoto.r.k@gmail.com

Tel/Fax: 092-802-5170

Reviewer #1: 

Comment1-1: I think Coding scheme (categorization of the participants’ responses) must be stated in detail. I understand that the participants put “rectangles” according to their interpretation of the communicator. This is a nice way to collect the participants’ interpretations because the participants can express their interpretation with more freedom than forced choice method. However, the categorization method sounds ambiguous.

Rectangles participants indicated may sometimes include more than one spot, such as one “rare” spot and one or two preceding spots (this may mean that “a change” occurred, from rare to not rare). How did you code such performance? I think it is necessary to describe the coding scheme completely in the text.

Reply1-1: The majority of participants’ response converged on a rare or a recent event. To prove this, we inserted new figures in supplementary materials (Figure S4 and Lines 324-329; Figure S5 and Lines 371- 372; Figure S6 and Lines 417 - 418) which shows the distribution of responses of each option. Specifically, in the Double-Rare-Events Condition, we believe that we should treat selecting only a latter rare event and selecting both rare events differently. It is because the aim is to explore the interaction of recency and rarity (Lines 374 - 376). In the UI Double-Rare-Events Condition, the question here is whether participants omit a rare event after the utterance from the candidate of referent (Lines 420 - 422). Thus, we treated selecting both rare events and selecting a rare event before utterance as different categories. We believe that our categorization scheme is an appropriate manner.

Tables S2 – S7 in supplementary materials shows that responses which cannot be categorized by any patterns. Some participants selected all options or against our prediction selected frequent events. We had not considered the possibility you suggested. We probed again responses categorized by unspecified strategies. However, the numbers of each response patters categorized by unspecified are too small. Additionally, these patters can reflect the possibility that participants remembered wrong sequences. Therefore, further exploration for these patterns is difficult. We inserted this explanation to reflect your comment (Page 11 in supplementary materials). We hope this satisfies your concerns.

Lines 324-329 in Results | Single-Rare-Event Condition with Adult participants

“Visualizing the distribution of responses confirms that the majority of responses converged on rare or recent events (Figure S4 for details). It seems reasonable that participants were largely categorized into either those who selected a rare event or those who selected a recent event (rare-event timing #-1). Those who used other strategies such as selecting all events equally were categorized as others (Table S2 – S7 for other strategies observed in the current study).”

Lines 371 - 372 in Results | Double-Rare-Events Condition with Adult participants

“The participants responses converged on selecting one or two rare events or selecting a recent event (Figure S5 for details).”

Lines 374 – 376 in Results | Double-Rare-Events Condition with Adult participants

“Selecting the 2nd rare event and selecting both rare events were treated as separate categories because the question here is to examine the interaction of recency and rarity”

Lines 417 – 418 in Results| UI Double-Rare-Events Condition with Adult

“The participants response converged on choosing one or more rare events or choosing a recent event (Figure S6 for details).”

Lines 420 – 422 in Results| UI Double-Rare-Events Condition with Adult participants

“The question here was whether participants omit a rare event after the utterance as a candidate for a referent. Thus, selecting both rare events and selecting only a rare event before the utterance were treated differently.”

Page 11 in Supplementary material

“The participants' responses were categorized based on the assumption that participants relied on a strategy to make decisions. It is also possible that the participants remembered the sequence incorrectly or chose options without any specific strategy, so the following categorization should remain speculative.”

Comment1-2: The results indicate that children treated recency and rarity cues separately, while adults treated recency as a continuous factor and integrated the gradient of recency with rarity. I’m not quite convinced about the authors’ statement. How do you know about being “separate” and being “continuous”? You seem to state it based on the AIC results, but still not very clear to me. More explanation is needed.

Reply1-2: Sorry for the lack of clarity. We were also confused and used the word of 'separate,' but your comments made us realize that it is unclear whether children separated or integrated rarity and recency only when recency peaked. Selecting a rare event when one occurred at #9 means choosing both a recent event and a rare event simultaneously. Thus, there are three possibilities: participants selected it as a rare event, as a recent event, or the interaction of recency and rarity mutually strengthened each other.

In both adults and children, the number of participants who selected a rare/recent event when a rare event occurred at #-1 is close to the sum of the participants who selected a rare or recent event when a rare event occurred at either #-3 or #-2. Thus, the highest number in #-1 might reflect the sum of participants who selected either a recent or rare event. However, adults showed linear increase between #-3 and #-2. This suggests adult treated recency as a continuous factor when integrating with rarity (Lines 584 - 591).

Children showed a nearly identical frequency of selecting a rare event between #-3 and #-2, with a noticeable increase only in #-1. This increase in #-1 might reflect the sum of participants who selected either a recent or rare event. If this possibility were dominant, they separated recency and rarity. However, another possibility is that they integrated recency and rarity only when recency peaked. Initially, we considered only the former possibility and thus argued “children treated recency and rarity separately”, but we have now included both possibilities in the current version of the manuscript (Lines 344 – 347, Lines 587 – 590 and Lines 631 - 632).

Children showed a nearly identical frequency of selecting a rare event between #-3 and #-2, with a noticeable increase only in #-1. This increase in #-1 might reflect the sum of participants who selected either a recent or rare event. If this possibility were dominant, they separated recency and rarity. However, another possibility is that they integrated recency and rarity only when recency peaked. Initially, we considered only the former possibility and thus argued “children treated recency and rarity separately”, but we have now included both possibilities in the current version of the manuscript (Lines 345 – 348, Lines 584 – 591 and Lines 632 - 633).

Except for the Double-Rare-Events Condition in adults, the number of times a rare event was selected when a rare event occurred at #-1 is close to the sum of the selections of either rare or recent events when a rare event occurred at #-2 and #-3 (Lines 582- 584; Lines 628 - 633). We assumed that the unique pattern in the Double-Rare-Events Condition is due to the stronger effect of rare events, leading some participants to choose both rare events (Lines 606-610). We now modified the sentence to reflect your comments.

Lines 345 – 348 in Results | Single-Rare-Event Condition with Adult participants

“At this point, it was unclear whether participants used the interaction of rarity and recency. However, it was unclear whether they used these two factors in a separate manner or integrated them only when recency and rarity overlapped."

Lines 582 – 584 in Implications from comparing adults and children

“The number of participants who selected a rare/recent event when a rare event occurred at #-1 is close to the sum of the participants who selected a rare or recent event when a rare event occurred at either #-3 or #-2.”

Lines 584 – 591 in Implications from comparing adults and children

“the fact that the highest percentage of participants selected a rare/recent event at #-1 might indicate that participants did not integrate recency and rarity and treated them separately. However, adults showed a gradual increase between #-3 and #-2, suggesting that they treated recency as a continuous factor when integrating with rarity. However, a limitation of our study is that it remains unclear whether children separated recency and rarity as categorical / stable factors, or whether they integrated them only when recency peaked.”

Lines 606 – 610 in Implications from comparing adults and children

“The number of adults who selected a rare/recent event when a rare event occurred at #-1 is lower than the total number of participants who selected a rare or recent event when a rare event occurred at either #-3 or #-2. This could be attributed to the fact that presenting two rare events had a stronger impact, leading some adults to choose both rare events.”

Lines 628 – 633 in Implications from comparing adults and children

“Again, the number of participants who selected a rare event when a rare event occurred at #-1 is close to the sum of the participants who selected a rare or recent event when a rare event occurred at either #-3 or #-2. Adults showed a linear increase as a function of position of the rare event. Thus, adults integrated rarity and recency.”

Lines 632 – 633 in Implications from comparing adults and children

“Children again treated recency either as categorical and separate or integrated them only when recency peaked.”

Trend analysis tested two hypotheses generated by our visual inspection. (1) Linear model assumes linear increase of selecting a rare event as a function of the position of a rare event. Thus, we replaced the position of a rare events #-3, #-2 and #-1 into -1, 0, 1, respectively. (2) Step model assumes a bounce of selecting a rare event when a rare event happened at #-1. Thus, we replaced the position of a rare events #-3, #-2 and #-1 into -0.5, -0.5, 1, respectively. These coefficients mean that #-3 and #-2 are the same value and only #-1 is higher than others. Lower AIC indicates better fitting of a model. Results showed that adults fitted to liner model better while children fitted to step model better along our hypothesis. To help readers understand this concept better, we now inserted detailed explanation of trend analysis (Lines 557 - 562) and modified the figure (see Figure 4). We hope this modification is your satisfactory. 

Lines 557 – 562 in Implications from comparing adults and children

“When a direction is clearly hypothesized, it can be expressed as a contrast [28]. This is the case for the current study because it also explored hypotheses generated through visual inspection. If a hypothesis predicts a particular pattern, this pattern can be represented numerically by assigning specific values to each condition. These assigned values, or contrast weights, capture the hypothesized differences in the means across experimental conditions.”

Comments 1-3: UII Double-Rare-Events

What does UII stand for? Could you tell me what kind of abbreviation (if it is an abbreviation) it is?

Reply1-3: We apologize for the confusion. We forgot to explain the abbreviation, and have now added the explanation. In response to a native speaker’s advice, UII Double-Rare-Events Condition was replaced into UI Double-Rare-Events Condition (Utterance In-between Double-Rare-Events Condition).

Lines 256 - 257

“Utterance In-between Double-Rare-Events Condition (hereafter UI Double-Rare-Events Condition)”

Comments 1-4: Sometimes one sentence is too long and difficult to read. Please state the content more

compact and concisely. For example,

The percentage of choosing both

404 1st and 2nd rare events (which is equivalent to rare events before and after the

405 utterance in the UII Double-Rare-Events Condition) decreased in the UII Double-Rare-

406 Events Condition compared to the Double-Rare-Events Condition (95 of 324 (29.32%)

407 in Double-Rare-Events Condition vs. 31 of 318 (9.75%) in UII Double-Rare-Events

Condition, χ2(1) 408 = 40.55, p < .001, Odds ratio = 0.26), suggesting that adults recognized

409 the difference between these two conditions and employed distinct strategies for each.

Reply1-4: We now corrected the sentence you pointed out (Lines 425 – 430). We again read our manuscript thoroughly, we shortened several sentences as follows (Lines 598 – 601; Lines 645 – 647; Lines 776 - 779).

Lines 425 – 430 in Results| UI Double-Rare-Events Condition with Adult participants

“The percentage of choosing both 1st and 2nd rare events (which is equivalent to rare events before and after the utterance in the UI Double-Rare-Events Condition) decreased in the UI Double-Rare-Events Condition compared to the Double-Rare-Events Condition (95 of 324 (29.32%) in Double-Rare-Events Condition vs. 31 of 318 (9.75%) in UI Double-Rare-Events Condition, χ2(1) = 40.55, p < .001, Odds ratio = 0.26).”

Lines 598 – 601 in Implications from comparing adults and children

“Children more frequently selected a recent event than adults did. This indicated that children’s interpretation of the utterance was inclined to recency when the relative frequency of rare events increased to 2 / 9 from 1 / 9 (in the Single-Rare-Event Condition) and the effect of rarity was thus weakened.”

Lines 645 – 647 in Discussion

“Across the three conditions, adults showed a linear increase in selecting a rare event as it approached the utterance. In contrast, children showed a bounce in selecting a rare event when a rare event occurred right before the utterance.”

Lines 776 – 779 in General Discussion

“Specifically, adults might perceive "Did you see that?" as a polysemous utterance that varies depending on the communicator’s interest, focus, perspective, experience or belief. In contrast , children interpret the same phrase more semantically, in other words, in a more literal manner.”

Comments 1-5: In the Supplementary material “… that two read arrows were overlaid” -> should be “… that two red arrows were overlaid”

Reply1-5: Thank you for pointing out. We now corrected the sentence you pointed out. 

Comments 1-6: 446the percentage of people 　　-> the percentage of children　

In the text, sometimes “children” and “people” seem to be mixed. “Children” sounds better.

Reply1-6: We reflect your comments by replacing people into “children” or participants for adults. In addition to your pointed here, we made this reflection through entire manuscript (Lines 465-469; Lines 477 - 482).

Lines 465 – 469 in Results | Single-Rare-Event Condition with Child participants

“The same analysis as used……, which posits that the percentage of children that chose rare events is not different from those that chose recent events. …… The results showed that the percentage of children that chose rare events significantly higher from that of recent events……”

Lines 477 – 482 in Results | Double-Rare-Events Condition with 

---

## [Decision Letter · Decision Letter 1]

13 Dec 2024

PONE-D-24-35473R1Recency and rarity effects in disambiguating the focus of utterance: A developmental studyPLOS ONE

Dear Dr. Kishimoto,

Thank you for submitting your manuscript to PLOS ONE. After careful consideration, we feel that it has merit but does not fully meet PLOS ONE’s publication criteria as it currently stands. Therefore, we invite you to submit a revised version of the manuscript that addresses the points raised during the review process.

I thank the authors for their attention to the reviewers' feedback.  After reviewing the revised manuscript, R1 raises a few additional minor points that should be addressed prior to acceptance.  Please address these comments and I will render a decision without sending the manuscript out for another round of peer review.

We look forward to receiving your revised manuscript.

Kind regards,

Laura Morett

Academic Editor

PLOS ONE

Journal Requirements:

Reviewers' comments:

Reviewer's Responses to Questions

**Comments to the Author**

1. If the authors have adequately addressed your comments raised in a previous round of review and you feel that this manuscript is now acceptable for publication, you may indicate that here to bypass the “Comments to the Author” section, enter your conflict of interest statement in the “Confidential to Editor” section, and submit your "Accept" recommendation.

Reviewer #1: All comments have been addressed

2. Is the manuscript technically sound, and do the data support the conclusions?

Reviewer #1: Yes

3. Has the statistical analysis been performed appropriately and rigorously? 

Reviewer #1: Yes

4. Have the authors made all data underlying the findings in their manuscript fully available?

Reviewer #1: Yes

5. Is the manuscript presented in an intelligible fashion and written in standard English?

Reviewer #1: Yes

6. Review Comments to the Author

Reviewer #1: The authors appropriately revised the manuscript. They responded and treated all questions I raised, in a sincere and thoughtful manner. I was impressed by the supporting information they provided in which the frequency and patterns of the “rare” type responses were shown.

There are some careless mistakes to be treated.

In the text

684 However, in this study, the dominantdominant cues available we

688 Considering the recency gradient and then integrating it with rarity requires

689 greater cognitive flexibility then treating both factors as categorical.

There are two “then(s).” May be the following sounds better with “,”.

688 Considering the recency gradient and then integrating it with rarity requires

689 greater cognitive flexibility, then treating both factors as categorical.

In adults, the P300 amplitudes in response to oddball stimuli increased as the

729 probability of standard/deviant stimuli increased; in other words, when oddball stimuli

730 became rarer. Infants as young as 6 months have shown sensitivity to deviant stimuli

731 [29], suggesting the attention to rare event in an epistemic state from early

732 development to adulthood. The stimuli in the current study had the same structure as in

45

the aforementioned oddball task (a monster who 733 played a different action was an

734 oddball).

->

This part is completely redundant. Please check the entire manuscript for redundancy.

In the Supporting information

Figure S5 The distributions of responses on each option in the UDouble-Rare-Events Condition

Should be UI.

7. PLOS authors have the option to publish the peer review history of their article (what does this mean?). If published, this will include your full peer review and any attached files.

Reviewer #1: **Yes: **Harumi Kobayashi

---

## [Author Response · Author response to Decision Letter 1]

19 Dec 2024

Dear Laura Morett

Dear professor Laura Morett

** November 2024

Thank you for inviting us to re-submit a revised version of our manuscript: “Recency and rarity effects in disambiguating the focus of utterance: A developmental study” (ID: PONE-D-24-35473) to PLOS ONE. We appreciate your time and effort as Editor and thank the reviewer for providing feedback. We have incorporated changes that reflect the detailed suggestions she has raised. We also hope that our edits and responses satisfy all the issues and concerns that she had. We have highlighted the revised parts in our manuscript and present a point-by-point response to the questions and comments below.

Sincerely,

Reiki Kishimoto

Faculty of Human Environment Studies

Kyushu University

E-A-306, 744 Motooka, Nishi-ku, Fukuoka

819-0395, Japan 

E-mail: kishimoto.r.k@gmail.com

Tel/Fax: 092-802-5170

Comment 1-1: 684 However, in this study, the dominantdominant cues available we

688 Considering the recency gradient and then integrating it with rarity requires

689 greater cognitive flexibility then treating both factors as categorical.

There are two “then(s).” May be the following sounds better with “,”.

In the Supporting information

Figure S5 The distributions of responses on each option in the UDouble-Rare-Events Condition

Should be UI.

Reply: Thank you for pointing out our typos. We replaced then into than to reflect your comment. As for the second part, we believe “than” sounds better. 

Lines 687 - 688

“However, in this study, the dominant cues available were their own salient cues, viewed from the communicator’s perspective.”

Lines 691 - 692

“Considering the recency gradient while integrating it with rarity requires greater cognitive flexibility than treating both factors as categorical.”

Page 25 in supplementary information

“Figure S5 The distributions of responses on each option in the UI Double-Rare-Events Condition.”

Comment 1-2:In the text

In adults, the P300 amplitudes in response to oddball stimuli increased as the

729 probability of standard/deviant stimuli increased; in other words, when oddball stimuli

730 became rarer. Infants as young as 6 months have shown sensitivity to deviant stimuli

731 [29], suggesting the attention to rare event in an epistemic state from early

732 development to adulthood. The stimuli in the current study had the same structure as in

45

the aforementioned oddball task (a monster who 733 played a different action was an

734 oddball).

->

This part is completely redundant. Please check the entire manuscript for redundancy.

Reply: You comments made me be aware that this paragraph repeated the same content. We edited this part to reflect your comments.

Lines 720 - 735

“In adults, P300 amplitudes in response to oddball stimuli increased as their probability decreased. Sensitivity to deviant stimuli has been observed in infants as young as 6 months [29], suggesting that attention to rare events develops early and persists into adulthood. The stimuli in the current study mirrored the oddball task structure, with a monster performing a different action serving as the oddball. Participants formed an expectation that the same event would recur, and when a rare event occurred, their expectation was violated, resulting in surprise. We developed our hypothesis based on the concept of entropy, surprise, memory, and causality. The results showed that participants across both age groups integrated these factors by attributing their perception to others. Children have already begun to understand that an element they perceive to be salient in a conversational context is likely to be perceived as salient by others as well. This belief allows them to pinpoint what is relevant for them from the communicator’s perspective. In the current paradigm, participants and their communicative partner observed novel events together. Participants had a belief that they shared the same knowledge and epistemic state with the communicative partner and that the partner was also aware of this belief.”

Comments 1-3: Please check the entire manuscript for redundancy.

Reply: We carefully reread our manuscript and realized that we frequently used multiple words in parallel with 'or' or 'and'. In most cases, however, the first word alone was sufficient to convey our ideas. Additionally, we often used unnecessary adjectives and adverb. Therefore, we omitted redundant words and edited some sentences in line with your comments. Additionally, we made several edits to address other redundant sentences.

We have already proceeded with the minor revisions. We acknowledge that some sentences still contain redundancy; however, we are concerned about altering the intended meaning. As a result, we made only minimal changes. We will take your advice into careful consideration for future work.

The deleted parts are marked with lines.

Lines 47 - 48

“Since any social species including humans cannot exchange their perspectives as long as they have independent minds, they rely on shared communicative rules [2, 3].”

Lines 53 - 55

“Mary attributes “that” to the most salient event, despite numerous other possibilities like the clouds, stars, or moon.”

Lines 59 - 60

“This fact means that receivers do not always succeed in decoding the communicator’s focus correctly.”

Lines 64 - 65

“Humans clarify referential ambiguities disambiguate information by discerning what information would a communicator perceive worthy from a receiver’s perspective.”

Lines 68 - 70

“As a result of this mutual process, the receiver attributes the communicator’s utterance to the relevant objects or events by … … [7].”

Lines 72 - 73

“They behave as if they were aware that the other individual sees, hears, perceives, processes or memorizes … ….”

Line 89

“Our focus here is how children calculate and/or integrate multiple factors.”

Lines 102 - 104

“Thus, their expectation is violated when a different event occurs [12]. Humans perceive consistency and expect a familiar event to recur when a series of repeated events is identical. When events occur infrequently, a surprising event stands out as new or unfamiliar.”

Lines 114 - 116

“This tendency increases over development [7, 15]. Humans are sensitive to rarity in addition to recency. The next question here is how children use other salient cues and whether these cues are outweighed by temporal proximity.”

Lines 131 - 133

“Another perspective supporting … … is the observation that animals tend to remember recent events more strongly and vividly [19].”

Lines 143 - 145

“The current study explored the development of pragmatic reasoning to identify … …what the communicator thinks is relevant for the receiver when they share perceptual information.”

Lines 171 - 174

“Additionally, … …, it is hypothesized that when a rare event occurred at any point during a sequence of events, and its timing varied, the degree of recency of this rare event increased… ….”

Lines 291 - 293

“The first three monsters performed … … a different event would be perceived as infrequent and rare.”

Lines 298 - 299

“They pressed a decision button on the upper left to proceed to the memory task, which tested whether participants correctly remembered when the utterance was presented.”

Lines 538 - 540

“The percentage of those … … to explore whether adults and children had the same inclination toward recency, or whether they differed.”

Lines 653 - 655

“The first (Single-Rare-Event) condition showed that, in the process of disambiguation, adults and children alike used both rarity and recency as cues but depended more on rarity.”

Lines 773 - 774

“Children’s interpretation of the demonstrative “that” might be restricted to the spatial or temporal regulation of the literal “that”.”

Lines 822 - 823

“they might discern that the communicator intends to share surprising or unexpected events that the communicator and receiver recently experienced together.”

---

## [Editor Report · Decision Letter 2]

26 Dec 2024

PONE-D-24-35473R2Recency and rarity effects in disambiguating the focus of utterance: A developmental studyPLOS ONE

Dear Dr. Kishimoto,

Thank you for submitting your manuscript to PLOS ONE. After careful consideration, we feel that it has merit but does not fully meet PLOS ONE’s publication criteria as it currently stands. Therefore, we invite you to submit a revised version of the manuscript that addresses the points raised during the review process.

We look forward to receiving your revised manuscript.

Kind regards,

Laura Morett

Academic Editor

PLOS ONE

Journal Requirements:

**Additional Editor Comments:**

I thank the authors for their attention to the remaining comments from R1. While I am satisfied with their responses, I noticed several additional language issues needing to be addressed prior to acceptance, which I list below [page.line]. I ask that the authors address them, and I will re-review the manuscript and render a decision once this is done.

[2.26] answered -> identified

[3.34] utterance -> utterances

[3.36] Delete "the" before "reasoning"

[4.43] Delete "the" before "information"

[4.46] Add "information" after "what;" delete "of the information"

[4.49] Delete "the" before "referential assignment;" delete "a" before "pragmatic"

[4.50-4.51] Delete "into;" delete "the" before "communicative" replace "which" with "that"

[4.53] Add "is" before "passing"

[5.58-5.59] the what makes Peter refer -> what Peter is referring to

[5.62] Delete "The" at beginning of sentence; replace "the one" with "that"

[5.63] Replace "which" with "that"

[5.65] would a communicator -> a communicator would

[5.68] shoe -> shoes

[5.71] have a belief -> believe

[6.79] Delete "each"

[6.80] the typical hearing -> individuals with typical hearing

[6.84] no additional -> minimal

[7.105] Add "is" before "consequently"

[8.112] which -> that

[8.118] Add "that" after "states"

[12.183] perspectives -> perspective

[15.237] University's -> University; staffs -> staff

[20.316] by R core team -> in R (add citation for R core team); add "the" after "with"

[21.332] Add "was" before "significantly;" from -> than

[23.367] participants -> participants'

[24.377] Add "was" before "significantly;" from -> than

[24.382] Add "was" before "significantly;" from -> than

[26.436-26.437] Add "was" before "significantly;" from -> than

[28.451] less -> fewer

[29.462] Delete "as" before "used"

[33.526] Add "a" before "strategy"

[33.530] standing -> considering

[33.541] Add "the" after "with"

[34.548] Add "the" after "in"

[34.551] Delete "onward;" insert comma after "children"

[35.576] Add "of" after "in favor"

[38.615] Evens -> Events

[40.543] bounce -> sharp increase

[40.645] which -> that

[40.651] the -> a

[41.663] which -> that

[41.666] bounce -> sharp increase

[42.687] Delete "semantically"

[42.692] And -> Moreover,

---

## [Author Response · Author response to Decision Letter 2]

26 Dec 2024

Reply: We have carefully re-checked all the typographical corrections you pointed out and confirmed their accuracy. The corresponding sentences have been revised accordingly in response to your suggestions.

---

## [Editor Report · Decision Letter 3]

29 Dec 2024

Recency and rarity effects in disambiguating the focus of utterance: A developmental study

PONE-D-24-35473R3

Dear Dr. Kishimoto,

We’re pleased to inform you that your manuscript has been judged scientifically suitable for publication and will be formally accepted for publication once it meets all outstanding technical requirements.

Kind regards,

Laura Morett

Academic Editor

PLOS ONE

Additional Editor Comments (optional):

I thank the authors for making the suggested language edits. This manuscript can now be accepted for publication in PLOS One.
---

## [Editor Report · Acceptance letter]

30 Jan 2025

PONE-D-24-35473R3 

PLOS ONE

Dear Dr. Kishimoto, 

I'm pleased to inform you that your manuscript has been deemed suitable for publication in PLOS ONE. Congratulations! Your manuscript is now being handed over to our production team.

Kind regards, 

on behalf of

Dr. Laura Morett 

Academic Editor

PLOS ONE